**Data Availability Statement:** The data are all contained within the manuscript.

**Funding:** The work was supported by the key projects of Sichuan circular economy research center, China, "Research on the countermeasures to improve the technological innovation efficiency

# Application of improved ELM algorithm in the prediction of earthquake casualties

**Xing Huang**[1], **Mengjie Luo**[1]*, **Huidong Jin**[2]

**1** School of Economics and Management, Southwest University of Science and Technology, Mianyang, China, **2** CSIRO Data61, Canberra, Australia

* lmj1714981490@qq.com

## Abstract

### Background

Earthquake casualties prediction is a basic work of the emergency response. Traditional forecasting methods have strict requirements on sample data and lots of parameters are required to be set manually, which can result in poor results with low prediction accuracy and slow learning speed.

### Method

In this paper, the Extreme Leaning Machine (ELM) is introduced into the earthquake disaster casualty predictions with the purpose of improving the prediction accuracy. However, traditional ELM model still has the problems of poor network structure stability and low prediction accuracy. So an Adaptive Chaos Particle Swarm Optimization (ACPSO) is proposed to the optimize traditional ELM's network parameters to enhance network stability and prediction accuracy, and the improved ELM model is applied to earthquake disaster casualty prediction.

### Results

The experimental results show that the earthquake disaster casualty prediction model based on ACPSO-ELM algorithm has better stability and prediction accuracy.

### Conclusion

ACPSO-ELM algorithm has better practicality and generalization in earthquake disaster casualty prediction.

## Introduction

Earthquake casualties prediction is a basic work for emergency rescue. The prediction of earthquake casualties can provide a basis for the emergency materials raising decisions of the emergency management department. At present, China's earthquake emergency response

of Mianyang enterprises under the effect of environmental regulation"(Grand No. XHJJ-1809) and Sichuan information management and service center, China, "Disaster risk assessment and demand identification of coastal cities based on multi-source data mining"(Grand No. SCXX2019ZD02).

**Competing interests:** The authors have declared that no competing interests exist.

capability has been obviously improved, but the collection work of emergency materials needs continuous improvement. The main performance is that the quantity and type of emergency materials cannot be effectively determined, which leads to insufficient or excessive emergency materials collection and increases obstacles to emergency rescue work. One of the most important reasons attributes to the failure of the effective prediction with casualties. So, after the earthquake, how to predict the casualties in disaster areas scientifically and effectively is a hot issue that many scholars have been studying.

Now the related research mainly focuses on two aspects: The construction of the index system of earthquake casualties prediction and the research of prediction methods. The research on the construction of earthquake casualties prediction index system, most of the results are based on disaster risk theory, and the index system constructed around the material attribute and social attribute of the index. For example, Murakami H O regarded earthquake intensity, staff occupancy rate and building collapse rate as the influencing factors of earthquake casualties [1]; Okada S divided buildings into different levels and predicts the number of casualties according to the damage degree of each level of the buildings [2]; Wen B C et al took the earthquake occurrence time, building earthquake-resistant level, magnitude and population density as the prediction indexes of casualties [3]; Samardjieva E et al studied the relationship between earthquake casualties and magnitude and population density [4]; Huang X et al proposed 11 earthquake casualty prediction indexes, including earthquake intensity, population density and the supply capacity of emergency materials [5]; Coburn A W took the staff occupancy rate, secondary disasters, building collapse rate and non-structural damage level as prediction indexes [6]; Zhang Jie and others built a prediction model for earthquake casualties based on the damaged area of houses [7]; Ma Yuhong and others put forward a prediction index system around environmental factors, prevention level, probability of secondary disasters and earthquake factors [8]. Now researchers at home and abroad have proposed a variety of methods for earthquake casualties prediction, including regression prediction model, uncertainty prediction model and machine learning method. For example, Ma Hongyan and others have established a grey prediction model for earthquake casualties in densely populated areas [9]; Muhammet G et al have constructed an artificial neural network prediction model and verified it with earthquakes of level 5 or above in Turkey [10]; Huang X et al proposed an earthquake death prediction model based on modified partial gaussian curve [11]; Gao Huiying et al established a rapid evaluation method for earthquake casualties by linear regression analysis [12]; Qian Fenglin et al. established an artificial neural network model for earthquake casualties prediction [13]; Ren Ningning et al. used rough set theory to reduce the prediction index structure and established a least square SVM prediction model to predict earthquake casualties [14].

According to the literature, existing research has come up with a relatively completed earthquake casualties prediction index system from different angles, but the representativeness and comprehensiveness of the index are still insufficient. Most of the literatures only consider the impact of single consistent disaster factor on casualties, and seldom extract casualty evaluation index from the perspective of disaster chain, resulting in a large difference between the evaluation results and the actual situation. In the research of prediction methods, regression prediction model focuses on the linear relationship between few indicators and casualties. Due to limited sample data, it is difficult to obtain a curve with high prediction accuracy, and regression prediction model is difficult to solve the prediction problem of high-dimensional indicators and non-linear indicators; the uncertain prediction model considers the small sample characteristics of seismic data and the uncertainty of indicators, but the data acquisition of some uncertain indicators is subjective. In order to solve the deficiencies of the regression prediction model and the uncertain prediction model, some researchers have introduced the machine learning method into the earthquake casualty prediction, which solves the high-

dimensional and non-linear problems of the prediction indicators well in certain extent, and improves the prediction accuracy. However, traditional machine learning methods lack certain fault tolerance and dynamics when dealing with complex factors, and lack certain timeliness and accuracy when dealing with massive data. Moreover, the parameters in traditional machine learning methods need manual adjustment, which leads to poor generalization ability and prone to over-fitting; therefore, it is necessary to seek a new machine learning method to improve the accuracy of earthquake casualties prediction. In recent years, some scholars have introduced the extreme learning machine (ELM) method into the prediction method. For example, Liu Caixia and others have used ELM method to predict the passenger volume of Chinese railways [15]. The results show that the prediction effect is remarkable, which provides an idea for this paper to study the prediction method of earthquake casualties. The core idea of ELM is to randomly select the input weights and hidden layer offsets of the network and keep the same in the training process. Compared with other traditional machine learning methods, ELM has obvious advantages such as simple implementation, fast learning speed and less human intervention. However, the traditional ELM method has disadvantages of low classification accuracy and poor stability of the network structure. Therefore, the prediction results are affected by network connection parameters and the number of hidden layer nodes.

For that reason, based on the theory of regional disaster system, and centering on the formation mechanism of disasters, starting from the interrelation among disaster-causing factors, disaster-pregnant environment and disaster-bearing bodies, this paper puts forward a highly representative and popularized earthquake casualty prediction index system; in the research of prediction method, ELM is introduced into the construction of earthquake casualty prediction method, an adaptive chaotic particle swarm optimization (ACPSO) algorithm is used to improve the traditional ELM, and the connection parameters of ELM network are optimized, so as to enhance the stability of ELM network and further improve the prediction accuracy of ELM.

## Earthquake disaster casualty prediction indexes

The Regional Disaster System Theory states that disasters are the result of interaction of disaster-causing factors, disaster-preparing environment, and disaster-bearing bodies. Disaster-causing factors refer to various natural and human factors that adversely affect human life, property or resources, such as drought, storm surge, frost, low temperature, hail, tsunamis, earthquakes, landslides, debris flow and so on, which are sufficient conditions for disaster formation; The disaster-bearing bodies refer to the main body of human society directly affected and damaged by the disaster, mainly including all aspects of human itself and social development, such as industry, people, agriculture, energy, construction, communication, various disaster reduction engineering facilities and production, life service facilities, and all kinds of wealth accumulated by people and so on, which is a necessary condition for disaster formation; The disaster-preparing environment is a comprehensive earth surface environment composed of the atmosphere, hydrosphere, lithosphere, biosphere, and human social circle. The sensitivity of the disaster environment provides a background for the interaction between disaster-causing factors and disaster-bearing bodies. According to the regional disaster system theory, the direct factor that causes the degree of casualties of earthquake casualties depends on the vulnerability of the disaster-bearing body. The greater the vulnerability of the disaster-bearing body is, the greater the casualty is. The vulnerability of the disaster-bearing body depends on the pregnancy environment, disaster-causing factors, and human resilience. When studying the prediction indexes of earthquake casualties, the four dimensions of disaster formation can be based on the theory of regional disaster system, disaster-causing factors, disaster-preventing

environment, disaster-bearing body and disaster-resisting ability. In the study of indexes, this paper firstly divides the influencing factors of earthquake disaster casualty prediction into four dimensions based on the regional disaster system theory, namely, the disaster-causing factor dimension, the disaster environment dimension, the disaster-bearing body dimension and the disaster resistance dimension; Secondly, around these four dimensions, three indexes of magnitude, epicentral intensity, and epicentral distance are used as secondary indexes of disaster-causing factors, and three indexes of earthquake occurrence time, earthquake geographical environment, and whether there are significant precursors are used as secondary indexes of disaster-causing environment, and the three indexes of population density, building fortification level, and damaged area of houses are used as secondary indicators of the disaster-bearing bodies; finally, using the primary component analysis method (PCA) to screen the primary selection indexes to determine the earthquake disaster prediction index system, as shown in Table 1.

## Casualty prediction model of earthquake disaster

### Traditional ELM prediction method

In view of the shortcomings of BP neural network, Professor Huang Guangbin proposed the concept of ELM, the ELM network lacks the output layer bias than the BP neural network. The input weight and hidden layer bias of the ELM network are generated randomly, and only need to determine the output weight, which can make up for the limitation of manual adjustment of the parameters of each layer in the BP neural network and improve the prediction accuracy. The ELM structure is shown in Fig 1.

In Fig 1, $x_1, x_2, \ldots, x_n$ are training sample data, $b_j$ is the threshold of the j-th neuron in the hidden layer, $w_{ij}$ is the connection weight of the node of the i-th input layer to the node of the j-th in the hidden layer, $O_L$ is the hidden layer node, $\beta_{jk}$ is the connection weight from the j-th hidden layer node to the k-th output layer node. Let the training sample set be $\{(x_i, y_i) | x_i \in R_n, y_i \in R_m, i = 1, 2, \ldots, N\}$, where $x_i = (x_{i1}, x_{i2}, \ldots, x_{in})$, $y_i = (y_{i1, y_{i2}}, \ldots, y_{im})$; The hidden layer $L$ is the number of neurons; Let the excitation function of the LEM network be $g(x)$. In this paper, Sigmoid is selected as the excitation function, $g(x)$ infinitely differentiable, then the ELM model can be expressed as,

$$\sum_{j=1}^{L} \beta_j g(w_j x + b_j) = y_l, l = 1, 2, \ldots, N \tag{1}$$

**Table 1. Earthquake casualties prediction indexes.**

| First-grade indexes | Second-grade indexes | Approach to data acquisition |
|---|---|---|
| Disaster-causing factors | Epicenter intensity | Subject to official Chinese reports |
| Disaster-pregnancy environment | Earthquake occurrence time | Subject to official Chinese reports |
| Disaster-bearing bodies | Damaged Area of Houses | Based on actual collapsed area |
| | Population density | Calculated according to the actual number of people per square kilometer, subject to Chinese official statistics |

Population density refers to the number of people per square kilometer. Building damage area refers to the total area of building collapse. Epicenter intensity refers to the intensity of the epicenter area, which is the highest intensity in an earthquake.

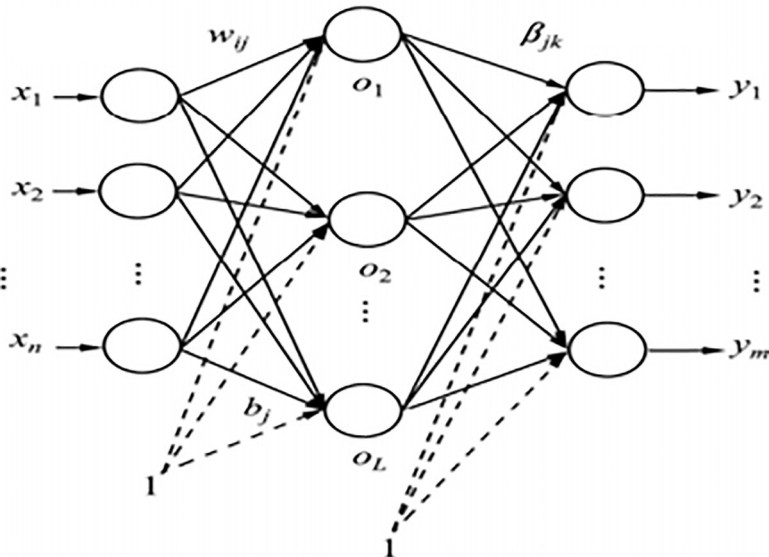

**Fig 1. ELM network topology.**

The matrix form is,

$$H\beta = Y \tag{2}$$

In formula (2), $\beta = [\beta_1, \beta_2, \ldots, \beta_L]^T_{l \times m}$, $Y = [y_1, y_2, \ldots, y_L]^T_{N \times M}$, $H = \begin{bmatrix} g(w_1 x_1 + b_1) & \cdots & g(w_L x_1 + b_L) \\ \cdot & & \cdot \\ \cdot & \cdots & \cdot \\ \cdot & & \cdot \\ g(w_1 x_N + b_1) & & g(w_L x_N + b_L) \end{bmatrix}_{N \times L}$;

Formula (2) is equivalent to solving the least squares solution of formula (3),

$$\hat{\beta} = \arg\min_{\beta} \|H\beta - Y\|_F \tag{3}$$

Formula (3) is solved as,

$$\hat{\beta} = H^+ T \tag{4}$$

In formula (4), $H^+$ is the generalized inverse of the hidden layer output matrix $H$.

## Using ACPSO algorithm to improve the traditional ELM

The biggest advantage of the ELM algorithm is that the connection weight between the input layer and the hidden layer, the threshold of the hidden layer can be set randomly, and no longer adjusted after setting, moreover, the connection weights between the hidden layer and the output layer do not need to be adjusted iteratively, but can be determined at once by solving the equations, through such rules, the generalization ability of ELM and training speed are greatly improved, but some parameters in the ELM model still need to be manually determined, and the ELM model randomly generates input weights and hidden layer thresholds. To a certain extent, this leads to unstable network structure and affects the prediction accuracy of the ELM model. In view of this, this paper will use the ACPSO algorithm to optimize the hidden layer parameters of ELM, optimize the input weights and hidden layer offsets randomly generated by ELM, and use the optimal weights and offsets as the input weights and hidden

Layer bias to enhance the stability of the ELM and the accuracy of the algorithm. In view of the instability of the ELM network structure, this paper will use ACPSO to improve the ELM algorithm, use the ACPSO algorithm to optimize the randomly generated M sets of input weights $\omega$ and hidden layer bias $b$, and use the optimal $\omega$ and $b$ as the ELM. input weight $\omega$ and hidden layer bias $b$, thereby enhancing the stability of the network and the accuracy of the algorithm.

Randomly generate $M$ sets of input weights $\omega$ and hidden layer bias $b$, and use each set of $\omega$, $b$ as the position vector of a particle in the particle swarm, namely $x_{td} = [\omega, b]$ (t = 1,2,$\cdots$,M, $d = 1,2,\cdots,D$, $D$ is the sum of the dimensions of $\omega$ and $b$).Using an iterative approach, each particle is brought closer to the best position it finds and the best particle in the group, so that the optimal solution of $\omega$, $b$ is searched. At each iteration, the particle updates the speed and position according to the following formulas.

$$v_{td}^{k+1} = \varpi \cdot v_{td}^{k} + c_1 r_1 (p_{td}^{k} - x_{td}^{k}) + c_2 r_2 (g_{td}^{k} - x_{td}^{k}) \tag{5}$$

$$x_{td}^{k+1} = x_{td}^{k} + v_{td}^{k+1} \tag{6}$$

$$\varpi = \varpi_{\max} - \frac{\varpi_{\max} - \varpi_{\min}}{K_{\max}} \times k \tag{7}$$

Among them: $v_{td} = [v_{t1}, v_{t2},\cdots,v_{tD}]$ is the flying speed of the particle t, that is the distance of the particle moves, the value range is $[v_{\min,d}, v_{\max,d}]$,this paper sets the particle velocity and range to [−1,1]; $c_1$ and $c_2$ represent learning factors, generally 2; $r_1$ and $r_2$ are random numbers between the interval [0,1]; The range of position $x_{td}$ is $[x_{\min,d}, x_{\max,d}]$; $P_{td}$ is the optimal position searched by the particle so far; $g_{td}$ is the optimal position searched by the entire particle swarm so far; $\varpi$ is the inertial weight, $\varpi_{\max}$ and $\varpi_{\min}$ are the maximum and minimum weight. The values in this paper are 0.9 and 0.4 respectively; $k$ is the current number of iterations, and $K_{\max}$ is the maximum number of iterations.

In the PSO algorithm, each particle represents a point with a certain speed, and each particle uses its corresponding individual fitness to judge the quality of the solution. In this paper, the simulation result error rate $f_t$ is used as the fitness value of network training. The smaller the $f_t$, the better the particle search performance.

$$f_t = \frac{err}{Q} \tag{8}$$

In the formula, $err$ is the number of errors in the simulation results; $Q$ is the total number of test samples. Since the initial particles are randomly generated, during the iteration process, when the particle position, individual extremum and group extremum are close, the speed update is determined by $\varpi \cdot v_{td}$. Since $\varpi < 1$, the particle speed becomes slower and slower, approaching zero, and the global search ability is lost, which eventually leads to a local minima. The chaotic search theory is introduced, and the randomness, regularity, and traversal performance of chaotic variables are used to effectively avoid particles from falling into local convergence during the optimization process. Add the chaotic variable $P_c$ to the optimal particle position variable, then:

$$x_{td}^{k+1} = (1 - \gamma)x_{td}^{k} + \gamma \cdot p_c \tag{9}$$

$$p_c = x_{\min,d} + Z_d^k(x_{\max,d} - x_{\min,d}) \tag{10}$$

$$\gamma^{k+1} = \lambda^{k+1}\left[1 - \left(\frac{k}{k+1}\right)^{\alpha}\right] \tag{11}$$

$$\lambda^{k+1} = \begin{cases} 0.9\lambda^k, m = 0 \\ 1, 0 < m \le 0.8M \\ 10\lambda^k, m > 0.8M \end{cases} \tag{12}$$

Among them: $P_c$ is the chaotic variable after the chaotic variable $z_d^k$ is normalized; $\gamma$ is the adaptive weight; $\alpha$ is the given constant; $m$ is the number of particles whose position is updated in the current iteration operation through the chaos search algorithm; $M$ is the total number of particles in the particle swarm; $z_d^k$ is the chaotic variable, generally generated by Logistic mapping:

$$z_d^{k+1} = \mu z_d^k(1 - z_d^k) \tag{13}$$

Among them: $\mu$ is the control parameter, $\mu \in (2,4)$, the initial value $z_d$ in each dimension ranges from [0,1]. When $\mu = 4$, the Logistic mapping is in a chaotic state, which can generate aperiodic and non-convergent chaotic variables.

The machine learning program based on the ACPSO-ELM algorithm is shown in Fig 2.

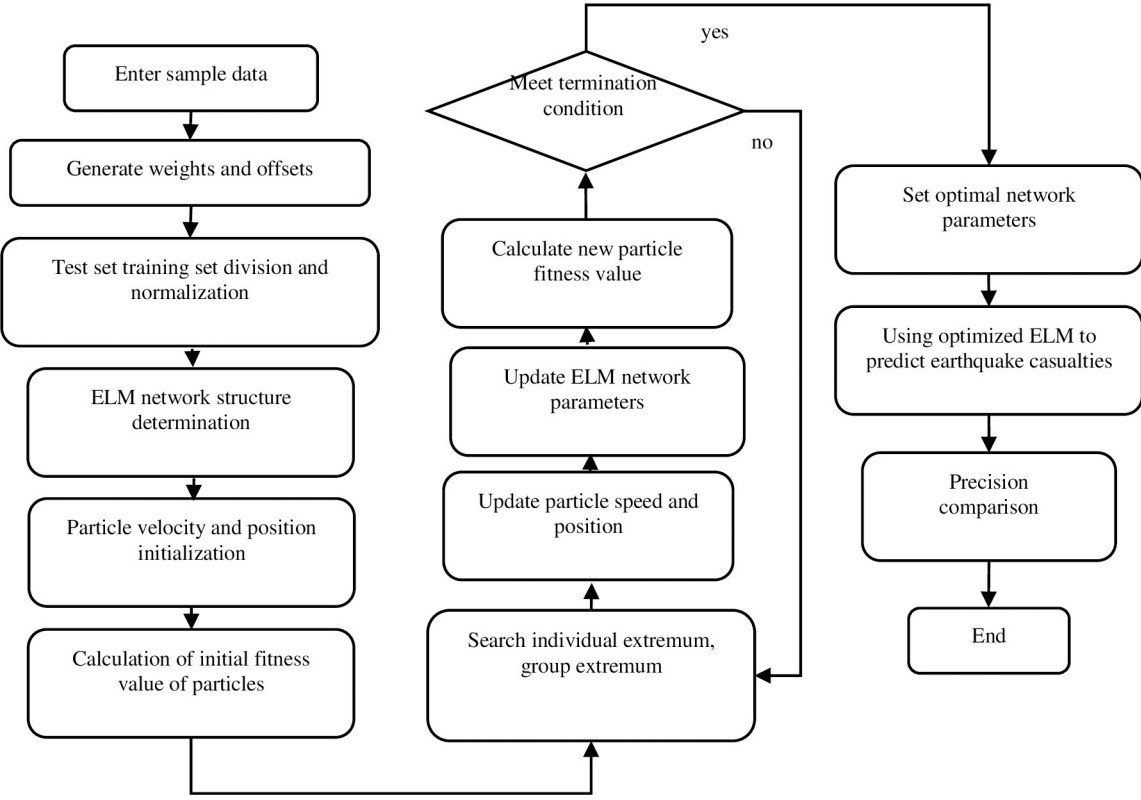

**Fig 2. Machine learning program based on the ACPSO-ELM algorithm.**

**Table 2. China's earthquake casualties from 1970 to 2017 [16].**

| Number | Place | Time of the earthquake | Epicentral intensity | Damaged area of house (1000$m^2$) | Population density(person/ $km^2$) | Earthquake casualties | |
|---|---|---|---|---|---|---|---|
| | | | | | | Dead | Injured |
| 1 | Lijiang, Yunnan | 19:14 | IX | 9590 | 66.67 | 309 | 17057 |
| 2 | Ninglang, Yunnan | 19:38 | VIII | 4011.4 | 36.68 | 5 | 1593 |
| 3 | Yaoan, Yunnan | 6:09 | VIII | 7380.048 | 114.48 | 7 | 2528 |
| 4 | Shidian, Yunnan | 11:13 | VIII | 403.255 | 161.8 | 3 | 235 |
| 5 | Wenchuan county of Sichuan | 14:28 | XI | 646252.110 | 13.3 | 87150 | 373643 |
| 6 | Tangshan, Hebei | 3:42 | XI | 16150 | 500 | 242000 | 164000 |
| 7 | Yingjiang, Yunnan | 8:24 | VIII | 54.915 | 45 | 5 | 130 |
| 8 | Yushu, Qinghai | 7:49 | IX | 9097.2 | 8.95 | 2968 | 12315 |
| 9 | Yaan, Sichuan | 8:02 | IX | 13815 | 98 | 217 | 13484 |
| 10 | Taiwan Strait | 14:20 | VIII | 14.2 | 236 | 3 | 671 |
| 11 | Puer, Yunnan | 5:34 | VII | 4476 | 55.8 | 3 | 562 |
| 12 | Jiashi, Xinjiang | 10:03 | IX | 2060 | 47.5 | 268 | 2058 |
| 13 | Nantou, Taiwan | 1:47 | XI | 1233.570 | 73.69 | 2378 | 8722 |
| 14 | Shangyi, Hebei | 11:52 | VIII | 6500 | 72.29 | 49 | 11439 |
| . | . | . | . | . | . | . | . |
| . | . | . | . | . | . | . | . |
| . | . | . | . | . | . | . | . |
| 72 | Songpan,Sicuan | 22:06 | IX | 7.5 | 17.5 | 41 | 756 |
| 73 | Inner Mongolia | 4:15 | VIII | 2171.160 | 34.84 | 28 | 865 |
| 74 | Haicheng, Liaoning | 19:36 | IX | 22400 | 1000 | 1328 | 16980 |
| 75 | Tonghai, Yunnan | 1:00 | X | 5076.840 | 267.03 | 15621 | 32431 |
| 76 | Minle, Gansu | 20:41 | VII | 904.090 | 63 | 10 | 46 |
| 77 | Yili, Xinjiang | 9:38 | VII | 120 | 39 | 10 | 47 |
| 78 | Heze, Shandong | 5:09 | VII | 5430 | 5.26 | 46 | 5138 |
| 79 | Yajiang, Sichuan | 8:09 | VIII | 332.12 | 4 | 3 | 55 |
| 80 | Yingjiang, Yunnan | 12:58 | VIII | 2041.30 | 69.25 | 25 | 314 |
| 81 | Minxian, Gansu | 7:45 | VIII | 3918 | 9.93 | 94 | 628 |
| 82 | Jingxian, Yunnan | 21:49 | VIII | 9228.6 | 38.64 | 1 | 324 |
| 83 | Kangding, Sichuan | 16:55 | VIII | 2576.2 | 9 | 5 | 54 |
| 84 | Jiuzhaigou, Sichuan | 21:19 | IX | 1105.065 | 1260 | 29 | 543 |

## The application of the model

### Data acquisition

The data of this paper came from literature [16], Evaluation Report of China Seismological Network, Earthquake Cases in China, China Statistical Yearbook, China Seismograph Network, and National Earthquake Data Center. The data includes 84 groups of sample data, as shown in Table 2.

Assuming input $x_i$ = {epicentral intensity, damaged area of house, earthquake occurrence time, population density}, and output $y$ = {the number of dead or injured},the experimental steps of earthquake casualties prediction based on the method of ACPSO-ELM are as follows

Step 1: Determine the training sample set, test data set and prediction data set. In this paper, ACPSO-ELM network will be trained with the maintenance method. Samples

Table 3. The accuracy of test in comparison of ACPSO, ELM and BP neural network.

| The average relative error and the average time consuming | ELM | | BP neural network | | ACPSO-ELM | |
|---|---|---|---|---|---|---|
| | The average relative error of death(%) | The average relative error of injured(%) | The average relative error of death(%) | The average relative error of injured(%) | The average relative error of death(%) | The average relative error of injured(%) |
| Average relative error(%) | 3.77 | 4.26 | 7.19 | 4.71 | 2.12 | 3.13 |
| Average time consuming(s) | 5.243 | 6.345 | 15.887 | 25.002 | 6.007 | 7.318 |

numbered 1~50 are selected as training samples of the model, samples numbered 51~71 are as test data of the model, and samples numbered 72~84 as predicted data of the model;

Step 2: Normalize the sample set to improve the rate of convergence of ACPSO-ELM algorithm;

Step 3: Establishing the three-layer ACPSO-ELM network structure with the neural network toolbox of Matlab R2016a, then determining the number of neurons in each layer and the excitation function, and selecting Sigmoid as the excitation function;

Step 4: ACPSO is used to optimize the parameters of ELM, and obtain the network optimal parameter values, including the optimal value of input layer to the hidden layer connected with weight $w$ and the hidden layer to the output layer connected with weight $\beta$ and the bias $b$;

Step 5: In order to compare the training effect of ACPSO-ELM with other methods, the traditional ELM model and BP neural network are involved in data training, then make comparisons of ACPSO-ELM, ELM model and BP neural network in the training accuracy;

Step 6: Inputting the sample data numbered 72~84 into the models of the ACPSO-ELM, ELM and BP neural network for the prediction of the earthquake casualty, and comparing their accuracy.

## Experimental process and results

**(1) Parameter determination.** The structure of ACPSO-ELM, ELM and BP neural network is: the number of input layer nodes is 4, the number of the hidden layer node is 3, the number of the output layer node is 2, the initial weight vector is uniformly distributed at (0, 1), and Sigmoid function is adapted as the excitation function of each layer; the learning rate of BP neural network is 0.05, the iteration is carried out by gradient descent method, it is finished when the learning accuracy reaches the minimum, and the initial threshold is given randomly.

**(2) Modal testing and validation.** 1) At first, the sample numbers 1~50 were put into modal training of ACPSO-ELM, ELM and BP neural network; then, and the sample numbers 51~71were put into model testing. The results are shown in Table 3.

As Table 3 shown, the test precision of ACPSO-ELM is the best, its average relative error of death is 2.12%, and the average relative error of injured is 3.13%; the training outcome of BP neural network is the worst, its average relative error of death is 7.19%, and the average relative error of injured is 4.71%; the test time for ELM is the least, its average time of death prediction is 5.243s, and the time of death injured prediction is 6.345s. The results of the earthquake death test are shown in Fig 3, and the results of the injury test are shown in Fig 4.

The *R-square* coefficient of ACPSO-ELM, ELM and BP neural network in Fig 3 is 0.95, 0.92 and 0.88, respectively. The ACPSO-ELM prediction model is the best, and its *R-square* coefficient is 0.95, indicating that the fitting effect of ACPSO-ELM prediction model is better than that of the traditional ELM and BP neural network.

In Fig 4, the *R-square* coefficient of ACPSO-ELM, ELM and BP neural network is 0.93, 0.88 and 0.84, respectively. The prediction effect of ACPSO-ELM model is the best, and its *R-square* coefficient is 0.93, indicating that the fitting effect of ACPSO-ELM prediction model is better

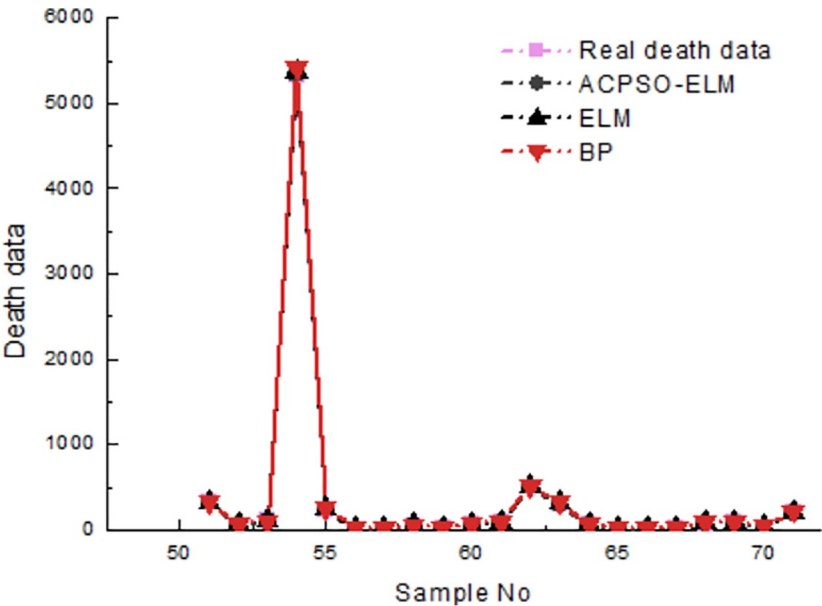

**Fig 3. The results of test in comparison of earthquake death.**

than that of the traditional ELM and BP neural network in the prediction of the injured in earthquake.

2) The samples numbered 72~84 were used as the prediction data, and they were respectively put into ACPSO-ELM, ELM and BP neural network for model prediction. The results are shown in Table 4.

Table 4 shows that ACPSO-ELM has the best predictive accuracy, its average relative error of death is 2.37% and the average relative error of injured is 3.02%; The predictive accuracy of

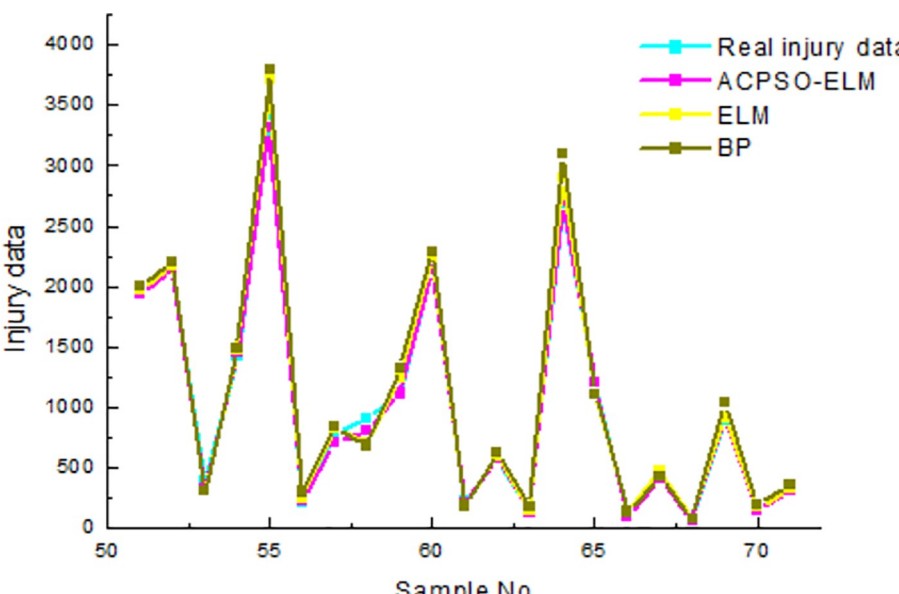

**Fig 4. The results of test in comparison of earthquake injured.**

**Table 4. The results of test in comparison of ACPSO, ELM and BP neural network.**

| The average relative error and the average time consuming | ELM | | BP neural network | | ACPSO-ELM | |
|---|---|---|---|---|---|---|
| | The average relative error of death(%) | The average relative error of injured(%) | The average relative error of death(%) | The average relative error of injured(%) | The average relative error of death(%) | The average relative error of injured(%) |
| Average relative error(%) | 3.51 | 4.33 | 7.43 | 4.33 | 2.37 | 3.02 |
| Average time consuming(s) | 3.123 | 4.109 | 9.432 | 14.308 | 4.112 | 5.098 |

BP neural network is the worst, its average relative error of death is 7.43% and the average relative error of injured is 4.33%. The prediction time of ELM is the least, its average time of death prediction is 3.123s, and its time of death injured prediction is 4.109s. The predicted result of earthquake death is shown in Fig 5, and the predicted result of the injured is shown in Fig 6.

In Fig 5, the *R-square* coefficient of ACPSO-ELM is 0.96, which is better than other prediction models, indicating that ACPSO-ELM has the best prediction effect on the number of earthquake death. The BP neural network has the worst prediction effect on the number of earthquake death, and its *R-square* is 0.86.

In Fig 6, the *R-square* coefficient of ACPSO-ELM is 0.91, which is better than other prediction models, indicating that ACPSO-ELM has the best prediction effect on the number of earthquake injured. The BP neural network has the worst prediction effect on the number of earthquake injured, and its *R-square* is 0.82.

In general, the ACPSO-ELM prediction model proposed in this paper has the advantages of good stability and high predictive accuracy.

## Conclusions

This paper uses the ACPSO algorithm has improved the traditional ELM model, to a certain extent, it improves the stability and prediction accuracy of the ELM network. The index of earthquake casualties prediction is proposed, and the four indexes of epicenter intensity, damaged area of house, earthquake occurrence time and population density are used as input

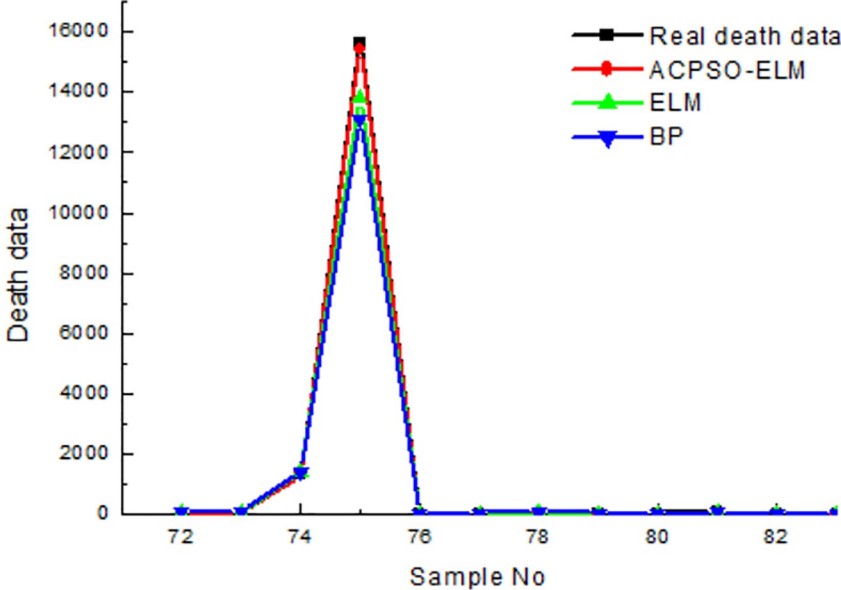

**Fig 5. The results of test in comparison of earthquake death.**

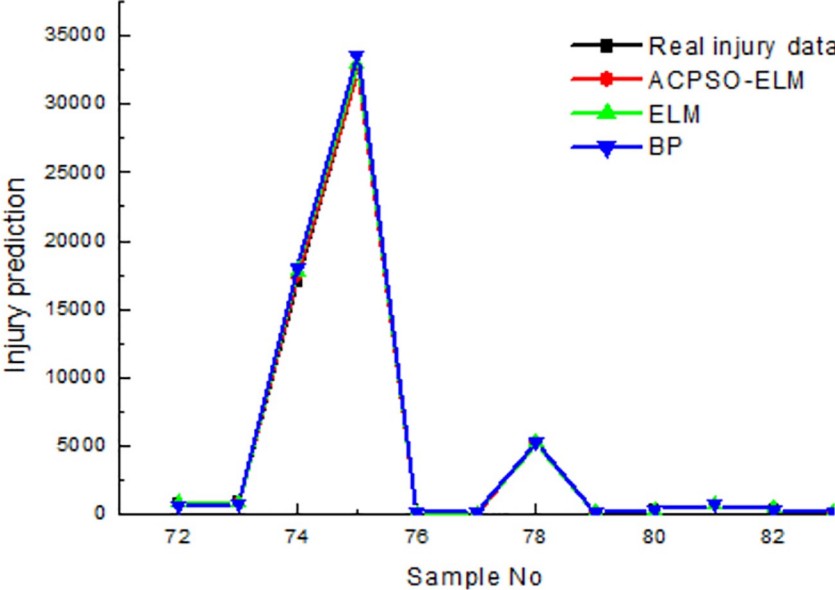

**Fig 6. The results of test in comparison of earthquake injured.**

indexes of ACPSO-ELM; the ELM model is improved by ACPSO algorithm, and the parameters of the ELM network are optimized to reduce the problems of network instability and low prediction accuracy caused by manually setting network parameters. The experimental results show that, compared with the traditional ELM model and BP neural network model, the extreme learning machine is used to predict the earthquake casualties. The ACPSO-ELM prediction method has the advantages of high stability and good prediction accuracy. This small sample data fitting provides a new reference, and provides a new method for earthquake casualty prediction.

## Acknowledgments

We wish to thank experts and journal editors who reviewed this article. We also wish to thank all scholars who provided references.

## Author Contributions

**Conceptualization:** Mengjie Luo.

**Investigation:** Xing Huang, Huidong Jin.

**Methodology:** Xing Huang, Huidong Jin.

**Resources:** Mengjie Luo.

**Writing – original draft:** Xing Huang, Mengjie Luo.

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
