## [Decision Letter · Decision Letter 0]

8 Apr 2020

PONE-D-20-05710

Casualty Prediction of Earthquake Disaster Based on Extreme Leaning Machine Algorithms

PLOS ONE

Dear Dr Huang,

Thank you for submitting your manuscript to PLOS ONE. After careful consideration, we feel that it has merit but does not fully meet PLOS ONE’s publication criteria as it currently stands. Therefore, we invite you to submit a revised version of the manuscript that addresses the points raised during the review process.

We would appreciate receiving your revised manuscript by May 23 2020 11:59PM. To enhance the reproducibility of your results, we recommend that if applicable you deposit your laboratory protocols in protocols.io, where a protocol can be assigned its own identifier (DOI) such that it can be cited independently in the future. For instructions see: http://journals.plos.org/plosone/s/submission-guidelines#loc-laboratory-protocols

We look forward to receiving your revised manuscript.

Kind regards,

Itamar Ashkenazi

Academic Editor

PLOS ONE

3. We noticed you have a minor occurrence of overlapping text with the following previous publication, on which you are an author, and which needs to be addressed:

"An earthquake casualty prediction model based on modified partial Gaussian curve" (https://doi.org/10.1007/s11069-018-3452-3)

In your revision ensure you cite all your sources (including your own works), and quote or rephrase any duplicated text outside the methods section. Further consideration is dependent on these concerns being addressed.

Reviewers' comments:

Reviewer's Responses to Questions

**Comments to the Author**

1. Is the manuscript technically sound, and do the data support the conclusions?

Reviewer #1: Yes

2. Has the statistical analysis been performed appropriately and rigorously? 

Reviewer #1: Yes

3. Have the authors made all data underlying the findings in their manuscript fully available?

Reviewer #1: Yes

4. Is the manuscript presented in an intelligible fashion and written in standard English?

Reviewer #1: No

5. Review Comments to the Author

Reviewer #1: In the introduction, from “Qian et al. selected magnitude … earthquake victims” needs revision.

There are many terms introduced in this section without giving the reader any idea of what they are such as “forecast level” or “personnel subsidence rate” “degree of secondary disasters”. An example or a very concise definition would help the readers to better understand the text.

I think the references should be followed by the year they were published. Please consult the journals website guide for authors to see if the citations have been properly formatted.

If the journal’s format allows, instead of a very long review of the factors related to the casualty prediction, the authors could tabulate them.

Extreme Learning Machine or E. Leaning Machine?

environment of pregnancy???

“Regional disaster system theory” should be capitalized if it’s a theory and cited.

disaster breeding surrounding???

In section 2.2.1., even though I believe that the terms used in this paper are not scientifically sound, but the authors didn’t mention which items are related to the elements at risk, risk, disaster etc.

“In this paper, nine indicators such as” why such as? Was there a longer list of items?

As for the PCA, there’s no need to provide the formula behind the method. As for the table, what are those ingredients (1 to 9). They need to be written in full or abbreviated if long. Plz also include the eigenvalues and explain why only 5 out of 9 indicators were selected as item 6 and 7 also seem to improve the total accuracy.

The authors didn’t mention anything such as earthquake level index before claiming to discard it. The basis on which the authors decided to delete this index is not clearly stated.

Before using the selected set of the indices, they need to be precisely defined and explained. They include epicenter intensity, building damage area, earthquake occurrence time and population density.

How did the authors reach the initial set of indicators? Just literature review, expert opinions, etc.? In the final set, plz state which item falls under which category of disaster risk factors, disaster breeding surrounding and disaster bearing substance

Is an 84 earthquake sample set is enough to trust the proposed method? Even though it has proved to be a very successful casualty prediction tool, but personally I believe that based on a limited sample we shouldn’t just rely on the numerical methods. What does the author think of this? Agreed or disagreed? Mention this in the paper.

The manuscript should be polished by a native speaker preferentially or someone with a good command of English grammar as I found multiple incomplete sentences and inconsistencies between subject and verb.

6. PLOS authors have the option to publish the peer review history of their article (what does this mean?). If published, this will include your full peer review and any attached files.

Reviewer #1: No

---

## [Author Response · Author response to Decision Letter 0]

18 May 2020

Author response

PONE-D-20-05710

Casualty Prediction of Earthquake Disaster Based on Extreme Leaning Machine Algorithms

 We noticed you have a minor occurrence of overlapping text with the following previous publication, on which you are an author, and which needs to be addressed: "An earthquake casualty prediction model based on modified partial Gaussian curve" (https://doi.org/10.1007/s11069-018-3452-3)。In your revision ensure you cite all your sources (including your own works), and quote or rephrase any duplicated text outside the methods section. Further consideration is dependent on these concerns being addressed.

Author response: Author noticed this paper has a minor occurrence of overlapping text with the previous publication, on which I am an author. The minor amount of repetition is mainly concentrated in the section of Introduction. The author has cited and modified it.

Reviewers' comments:

Reviewer's Responses to Questions

Comments to the Author

1. Is the manuscript technically sound, and do the data support the conclusions?

Reviewer #1: Yes

2. Has the statistical analysis been performed appropriately and rigorously? 

Reviewer #1: Yes

3. Have the authors made all data underlying the findings in their manuscript fully available?

Reviewer #1: Yes

4. Is the manuscript presented in an intelligible fashion and written in standard English?

Reviewer #1: No

5. Review Comments to the Author

Reviewer #1: In the introduction, from “Qian et al. selected magnitude … earthquake victims” needs revision.

Author response: author had revision, as follows: Qian Fenglin et al. established an artificial neural network model for earthquake casualties prediction [13].

There are many terms introduced in this section without giving the reader any idea of what they are such as “forecast level” or “personnel subsidence rate” “degree of secondary disasters”. An example or a very concise definition would help the readers to better understand the text.

Author response: author had given some definition to explain these terms. For example: Disaster-causing factors: refer to various natural and human factors that adversely affect human life, property or resources, such as drought, storm surge, frost, low temperature, hail, tsunamis, earthquakes, landslides, debris flow and so on. The disaster-bearing bodies refer to the main body of human society directly affected and damaged by the disaster, mainly including all aspects of human itself and social development, such as industry, people, agriculture, energy, construction, communication, various disaster reduction engineering facilities and production, life service facilities, and all kinds of wealth accumulated by people and so on. The disaster-preparing environment is a comprehensive earth surface environment composed of the atmosphere, hydrosphere, lithosphere, biosphere, and human social circle. 

I think the references should be followed by the year they were published. Please consult the journals website guide for authors to see if the citations have been properly formatted.

Author response: the author has modified the citation format, for example: Wen B C, Jiang C., 2013, Forecasting Emergency Demand Based on BP Neural Network and Principal Component Analysis. Adv Inform Serv Sci, 5(13):38-45.

Extreme Learning Machine or E. Leaning Machine?

Author response: In this paper, ELM is Extreme learning machine(ELM).

environment of pregnancy?

Author response: The author corrects this term as " disaster environment" and gives its definition, as follows: The sensitivity of the disaster environment provides a background for the interaction between disaster-causing factors and disaster-bearing bodies. 

 “Regional disaster system theory” should be capitalized if it’s a theory and cited.

Author response: author had modified as: The Regional Disaster System Theory.

disaster breeding surrounding?

Author response: author had revised the term as: disaster environment, and gave the define: disaster environment provides a background for the interaction between disaster-causing factors and disaster-bearing bodies. 

“In this paper, nine indicators such as” why such as? Was there a longer list of items?

Author response: The author added some contents to screen these 9 indexes, as follows: Secondly, around these four dimensions, three indexes of magnitude, epicentral intensity, and epicentral distance are used as secondary indexes of disaster-causing factors, and three indexes of earthquake occurrence time, earthquake geographical environment, and whether there are significant precursors are used as secondary indexes of disaster-causing environment, and the three indexes of population density, building fortification level, and damaged area of houses are used as secondary indicators of the disaster-bearing bodies; finally, using the primary component analysis method (PCA) to screen the primary selection indexes to determine the earthquake disaster prediction index system, as shown in Table 1.

Table 1 Earthquake Casualties Prediction Indexes 

First-grade indexes Second-grade indexes Approach to data acquisition 

Disaster-causing factors Epicenter intensity Subject to official Chinese reports

Disaster-pregnancy environment Earthquake occurrence time Subject to official Chinese reports

Disaster-bearing bodies Damaged Area of Houses Based on actual collapsed area

 Population density Calculated according to the actual number of people per square kilometer, subject to Chinese official statistics

As for the PCA, there’s no need to provide the formula behind the method. As for the table, what are those ingredients (1 to 9). They need to be written in full or abbreviated if long. Plz also include the eigenvalues and explain why only 5 out of 9 indicators were selected as item 6 and 7 also seem to improve the total accuracy.

Author response: The author has deleted this PCA formula and revised some contents to answer the expert's suggestions.

The Regional Disaster System Theory states that disasters are the result of interaction of disaster-causing factors, disaster-preparing environment, and disaster-bearing bodies. Disaster-causing factors refer to various natural and human factors that adversely affect human life, property or resources, such as drought, storm surge, frost, low temperature, hail, tsunamis, earthquakes, landslides, debris flow and so on, which are sufficient conditions for disaster formation; The disaster-bearing bodies refer to the main body of human society directly affected and damaged by the disaster, mainly including all aspects of human itself and social development, such as industry, people, agriculture, energy, construction, communication, various disaster reduction engineering facilities and production, life service facilities, and all kinds of wealth accumulated by people and so on, which is a necessary condition for disaster formation; The disaster-preparing environment is a comprehensive earth surface environment composed of the atmosphere, hydrosphere, lithosphere, biosphere, and human social circle. The sensitivity of the disaster environment provides a background for the interaction between disaster-causing factors and disaster-bearing bodies. According to the regional disaster system theory, the direct factor that causes the degree of casualties of earthquake casualties depends on the vulnerability of the disaster-bearing body. The greater the vulnerability of the disaster-bearing body is, the greater the casualty is. The vulnerability of the disaster-bearing body depends on the pregnancy environment, disaster-causing factors, and human resilience. When studying the prediction indexes of earthquake casualties, the four dimensions of disaster formation can be based on the theory of regional disaster system, disaster-causing factors, disaster-preventing environment, disaster-bearing body and disaster-resisting ability. In the study of indexes, this paper firstly divides the influencing factors of earthquake disaster casualty prediction into four dimensions based on the regional disaster system theory, namely, the disaster-causing factor dimension, the disaster environment dimension, the disaster-bearing body dimension and the disaster resistance dimension; Secondly, around these four dimensions, three indexes of magnitude, epicentral intensity, and epicentral distance are used as secondary indexes of disaster-causing factors, and three indexes of earthquake occurrence time, earthquake geographical environment, and whether there are significant precursors are used as secondary indexes of disaster-causing environment, and the three indexes of population density, building fortification level, and damaged area of houses are used as secondary indicators of the disaster-bearing bodies; finally, using the primary component analysis method (PCA) to screen the primary selection indexes to determine the earthquake disaster prediction index system, as shown in Table 1.

The authors didn’t mention anything such as earthquake level index before claiming to discard it. The basis on which the authors decided to delete this index is not clearly stated.

Author response: The author has modified the index selection part. Please read the above answer.

Before using the selected set of the indices, they need to be precisely defined and explained. They include epicenter intensity, building damage area, earthquake occurrence time and population density.

Author response: The author has defined these terms under table 1,as follows:

Note: Population density refers to the number of people per square kilometer. Building damage area refers to the total area of building collapse. Epicenter intensity refers to the intensity of the epicenter area, which is the highest intensity in an earthquake.

How did the authors reach the initial set of indicators? 

Author response: The ways of obtaining the initial indicators include theoretical analysis of disaster system, literature research and expert interviews.

Is an 84 earthquake sample set is enough to trust the proposed method? Even though it has proved to be a very successful casualty prediction tool, but personally I believe that based on a limited sample we shouldn’t just rely on the numerical methods. What does the author think of this? Agreed or disagreed? Mention this in the paper.

Author response: authors agree with the opinions of reviewer. In order to show the advantages of the model, the author improved the traditional ELM, and compared the model with BP neural network and traditional ELM.

The manuscript should be polished by a native speaker preferentially or someone with a good command of English grammar as I found multiple incomplete sentences and inconsistencies between subject and verb.

Author response: The manuscript had be polished by a native speaker preferentially, including English grammar, incomplete sentences and inconsistencies between subject and verb, and so on.

In addition, in order to improve the prediction accuracy of the model, the author improved the traditional elm. The results show that the improved elm model is better than the traditional elm model.

The author also revised the title of the paper, as "Application of improved ELM algorithm in the prediction of earthquake casualties".

---

## [Decision Letter · Decision Letter 1]

11 Jun 2020

Application of improved ELM algorithm in the prediction of earthquake casualties

PONE-D-20-05710R1

Dear Dr. Huang,

We’re pleased to inform you that your manuscript has been judged scientifically suitable for publication and will be formally accepted for publication once it meets all outstanding technical requirements.

Kind regards,

Itamar Ashkenazi

Academic Editor

PLOS ONE

Additional Editor Comments (optional):

Reviewers' comments:

Reviewer's Responses to Questions

**Comments to the Author**

1. If the authors have adequately addressed your comments raised in a previous round of review and you feel that this manuscript is now acceptable for publication, you may indicate that here to bypass the “Comments to the Author” section, enter your conflict of interest statement in the “Confidential to Editor” section, and submit your "Accept" recommendation.

Reviewer #1: All comments have been addressed

2. Is the manuscript technically sound, and do the data support the conclusions?

Reviewer #1: Yes

3. Has the statistical analysis been performed appropriately and rigorously? 

Reviewer #1: Yes

4. Have the authors made all data underlying the findings in their manuscript fully available?

Reviewer #1: Yes

5. Is the manuscript presented in an intelligible fashion and written in standard English?

Reviewer #1: Yes

6. Review Comments to the Author

Reviewer #1: Dear Editor

I have fully read the manuscript and believe the authors have carefully addressed all my concerns. Therefore I believe the paper is suitable for publication now.

Best Regards

7. PLOS authors have the option to publish the peer review history of their article (what does this mean?). If published, this will include your full peer review and any attached files.

Reviewer #1: No

---

## [Editor Report · Acceptance letter]

16 Jun 2020

PONE-D-20-05710R1 

Application of improved ELM algorithm in the prediction of earthquake casualties 

Dear Dr. Huang:

I'm pleased to inform you that your manuscript has been deemed suitable for publication in PLOS ONE. Congratulations! Your manuscript is now with our production department. 

Kind regards, 

on behalf of

Dr. Itamar Ashkenazi 

Academic Editor

PLOS ONE